# An Interesting Conversion Route of Mononuclear Zinc Complex to Zinc Mixed Carboxylate Coordination Polymer

**DOI:** 10.3390/molecules28052011

**Published:** 2023-02-21

**Authors:** Gina Vasile Scăețeanu, Constantin G. Daniliuc, Rodica Olar, Mihaela Badea

**Affiliations:** 1Department of Soil Sciences, University of Agronomical Sciences and Veterinary Medicine, 59 Mărăști Str., 011464 Bucharest, Romania; 2Organisch-Chemisches Institut, Westfälische Wilhelms-Universität Münster, Corrensstrasse 40, 48149 Münster, Germany; 3Department of Inorganic, Organic Chemistry, Biochemistry and Catalysis, Faculty of Chemistry, University of Bucharest, 90–92 Panduri Str., 050663 Bucharest, Romania

**Keywords:** acrylate, coordination polymer, formate, structure, zinc(II)

## Abstract

A complex [Zn(bpy)(acr)_2_]·H_2_O (**1**) was converted in a DMF medium (DMF = *N,N*′-dimethylformamide) into a coordination polymer [Zn(bpy)(acr)(HCOO)]_n_ (**1a**) (bpy = 2,2′-bipyridine, and Hacr = acrylic acid), and the species was fully characterized through a single crystal X-ray diffraction. Additional data were obtained by IR and thermogravimetric analysis. Complex (**1a**) crystalized the coordination polymer in the space group *Pca*2_1_ of an orthorhombic system. Structural characterization revealed that Zn(II) adopted a square pyramidal stereochemistry generated by bpy molecules, coordinated by chelate, acrylate, and formate ions as unidentate and bridged, respectively. The presence of both the formate and the acrylate with different coordination modes generated two bands in ranges that were characteristic for the carboxylate vibration modes. Thermal decomposition occurs in two complex steps: it first happens via a bpy release, which is followed by an overlapped process that is associated with acrylate and formate decomposition. The obtained complex is of present interest due to the presence of two different carboxylates in its composition and situation, which is rarely reported in the literature.

## 1. Introduction

Some metallic ions are recognized for their ability to catalyze the hydrolysis reaction of amidic bonds, with this category being framed as the following: Cu(II) [1,2], Co(III) [3], Ni(II) [4], Zn(II) [5], and Ln(III) (Eu, Gd, Tb, and Nd) [6,7].

For example, two linear trinuclear Cu(II) complexes of 3,5-pyrazoledicarboxylic acid (H_3_L) formulated [Cu_3_(L)_2_(Me_2_en)_2_(H_2_O)_2_]∙(H_2_O)_8_ and [Cu_3_(L)_2_(MeOH)_6_(H_2_O)_4_], where Me_2_en is *N,N*-dimethyl-ethylenediamine, have been obtained from a precursor ligand, through copper-induced amide hydrolysis, and this route is successful since the attempts to use raw materials as their components failed, leading to the same dinuclear complex, Na_2_[Cu_2_L_2_(H_2_O)_2_] [1].

Complexes with the above mentioned metallic ions that catalyze amidic bonds can be considered as models for metallopeptidases [2]. In addition, carboxypeptidase A is a Zn(II) metalloenzyme that catalyzes the hydrolysis of an amidic bond from peptides [5]. Amidic bonds of some peptides that contain threonine and histidine residues are hydrolyzed in the presence of Ni(II) ions [4].

In the coordination chemistry area, *N*,*N*′-dimethylformamide (DMF), a liquid representative member of the amide class, is used in various syntheses as the aprotic solvent [8,9], as the ligand that usually acts as unidentate O-donor [10,11,12,13], rarely bridging ligand [14], or as an uncoordinated crystallization molecule [13,15]. For example, coordination polymers with luminescent properties and general formula ^2^_∞_[Ln(tfBDC)(NO_3_)(DMF)_2_]∙DMF, where Ln = Eu^3+^, Gd^3+^, Tb^3+^, Ho^3+^, Tm^3+^, and tfBDC^2−^ = 2,3,5,6-tetrafluoroterephtalate, present into composition DMF molecules that act differently: ligand coordinated through O atom and crystallization molecule [15]. In the case of the tetranuclear complex [NiL(H_2_O)Pb(SCN)_2_(DMF)]_2_ (LH_2_ = N,N′-bis(salicylidene)-1,3-diaminopropane), Kurtaran and co-workers [14] reported that DMF acted as bridge between the Pb(II) ions.

In the literature, there are reported situations when DMF hydrolysis occurs, and after this process, the afforded complexes contain formate ions [2,6,16,17], dimethylamine [18,19,20] or dimethylammonium cations [16,20,21,22]. Occasionally, there are obtained complexes that include both the formate and dimethylamine [16].

Concerning the DMF hydrolysis process, there are many suppositions, but it seems that until now, a detailed mechanism is not clear. For instance, Kim and their co-workers [2] considered that the possible pathways of DMF hydrolysis in the case of a tetraazomacrocyclic copper (II) complex are either activations of amidic bonds by a carbonyl coordination of DMF to copper or nucleophilic attacks of hydroxo species that have resulted during the complex synthesis to DMF, leading to amidic bond cleavage. The formate ion acts as unidentate ligand and it coordinates with the copper ion, generating a monomeric complex [2].

Other authors [16] sustain the importance of an acidic medium (pH = 4) during DMF hydrolysis; this reaction condition leads to formate ions and dimethylamine. Furthermore, acidic hydrolysis may occur under normal or solvothermal conditions [6,16,23].

In addition, formate coordination polymers with dye adsorption properties were obtained by Zhu and coworkers [23], using in-situ hydrolysis of DMF under temperature and acidic medium.

Keggin heteropolyacid, H_3_[PMo_12_O_40_]·4H_2_O, catalyzes the hydrolysis of DMF at room temperature, and the formic acid that is obtained during hydrolysis is oxidized by 12-molybophosphate acid into carbon dioxide, leading the hydrolysis balance of the DMF to shift toward the generation of dimethylamine. The final obtained product is [(CH_3_)_2_NH_2_]_3_[PMo_12_O_40_] [18].

A 2D-coordination polymer, [Cu(bdc)(NHMe_2_)_2_], where H_2_bdc is 1,4-benzene dicarboxylic acid, was obtained through solvothermal synthesis when DMF hydrolysis was conducted unexpectedly to dimethylamine, according to the authors of [19], which coordinates to the copper center.

Decarbonylation of DMF upon heating under basic conditions led to dimethylammonium cations, which were identified as the composition of the complex [NH_2_Me_2_]_2_[Zn_3_(μ-bdc)_4_]∙DMF∙H_2_O [21].

Another interesting piece of research was reported by Tambornino and Hoch [17], which found that solutions in the DMF of complexes [Eu(DMF)_8_]I_3_ and [Nd(DMF)_8_]I_3_ afforded dodecanuclear species, [RE_12_(DMF)_24_ (HCOO)_8_(OH)_16_]I_3_·4DMF (RE = Eu, Nd) with formate ions resulting from DMF hydrolysis.

The results of the current research are based on the catalytic activity of Zn(II) from a mononuclear complex, [Zn(bpy)(acr)_2_]·H_2_O (**1**) (bpy = 2,2′-bipyridine, and Hacr = acrylic acid) [24] over DMF, which was converted to a formate ion during a hydrolysis reaction. The transformation occurred in the presence of melamine (MA), which was intended to act as a ligand; this was the initial objective of the approach. The formate ion that resulted after the hydrolysis acted as a bridging ligand, leading to a polynuclear complex, which was outside the expectations of our original purpose.

The literature survey related to mixed carboxylate complexes is scarce, but the research of the already published data has provided a few examples of such compounds [25,26,27,28,29,30,31,32] with various structures or properties that were obtained by different routes, including serendipitously. Therefore, an important aspect that is worth mentioning is that, so far, to our knowledge, complexes with acrylate and other carboxylate ligands have not been reported, and this complex is the first of its kind.

## 2. Results and Discussion

### 2.1. Synthesis of the Complex

In an attempt to coordinate melamine to [Zn(bpy)(acr)_2_]·H_2_O (**1**) in the DMF reaction medium, the complex [Zn(bpy)(acr)(HCOO)]_n_ (**1a**) was instead obtained as a result of the DMF undergoing a formate transformation during the hydrolysis (Figure 1). As a possible explanation of this unexpected result, we could suppose that Zn(II) ion acted as a catalyst of amidic bond hydrolysis, which is already reported by other authors [5]. Additionally, as DMF hydrolysis is influenced by the reaction conditions (acidic/basic medium, heating, etc.) [16,21,23], it could be assumed that melamine (MA) could also catalyze amidic bond formation by creating a basic medium.

### 2.2. X-ray Crystal Structure of Complex

A summary of the crystallographic data and the structure refinement of [Zn(bpy)(acr)(HCOO)]_n_ (**1a**) are given in Appendix A. Single crystal X-ray diffraction revealed a zig-zag polynuclear structure running along axis a, with the Zn(II) coordinated by the chelate ligand 2,2′-bipyridine, by two formate ligands in a μ^2^-κ^1^-O: κ^1^-O′ fashion and one acrylate ion as unidentate (Figure 1). The Zn(II) shows a five coordinate surrounding with a distorted square pyramidal geometry (τ = 0.17). The parameter τ is defined as (β − α)/60 [where β = N1—Zn1—O3, α = O1—Zn1—N2], and its value varies from zero (in regular square pyramidal) to one (in regular trigonal bipyramidal) [33,34]. It should be noted that the acrylate ion presents an asymmetric chelating mode of the carboxylate group O1/O2, as indicated by the Zn–O bond values (Zn1–O1 2.076 Å and Zn1–O2 2.562 Å), with a difference of ca. 0.48 Å, as was also observed for other Zn(II)-based carboxylate polymers [35,36]. Consequently, the Zn1–O2 bond can be considered only a weakly coordinated interaction. The bonds’ lengths of Zn–O and Zn–N involving the formate and bpy ligands are in the expected range (Zn1–O3 2.042 Å; Zn1–O4 2.022 Å; Zn1–N1 2.158 Å; and Zn1–N2 2.147 Å). The bond lengths of Zn-O in the case of the formate were shorter than those found in the case of the acrylate. The Zn-N bond lengths in the complex used as raw material [Zn(bpy)(acr)_2_]·H_2_O (**1**) [24] were slightly shorter than those identified in the resulting one [Zn(bpy)(acr)(HCOO)]_n_ (**1a**).

An analysis of the packing diagram reveals the formation of a 2D network along the a-axis involving π⋯π interactions between the bipyridine ligands (shortest distance between the aromatic rings 3.537 Å) supported by additional C-H⋯O interactions between the aromatic C-H units and formate and acrylate ligands (Figure 2 and Table 1).

### 2.3. IR Spectra

A comparison of the IR spectra of complexes (**1**) and (**1a**) revealed the similarities and differences observed in the range of the absorption bands that were characteristic of the bpy and carboxylate groups, respectively (Figure 3).

The most important absorption bands identified in these spectra are summarized in Table 2. In the range 1450–1500 cm^−1^, the characteristic bands of the chelate bpy arise. The presence of carboxylate anions in the composition of complexes is supported by the appearance of bands that can be assigned to the ν_as_(COO) and ν_s_(COO) vibration modes. According to the literature, the Δ value (Δ = ν_as_(COO) − ν_s_(COO)) can be associated with the coordination mode of the carboxylate group [37,38,39]. Therefore, for the complex (**1a**), a Δ value of 253 cm^−1^ resulting from bands located at 1619 and 1366 cm^−1^, indicates an unidentate coordination for acrylate, while the value of 188 cm^−1^ suggests a bridge bidentate coordination mode for formate [40], as was undoubtedly otherwise indicated by the single crystal X-ray data.

### 2.4. Thermal Behavior

The thermogravimetric measurement (TG) in the air showed (Figure 4) that the polymeric complex [Zn(bpy)(acr)(HCOO)]_n_ (**1a**) was thermally stable up to 240 °C, followed by its decomposition in two well-defined steps. In the first step, a mass loss of 47.1% corresponded to the bpy elimination (calculated mass loss: 46.2%), which was accompanied by an endothermic effect. In the second step, the decomposition of both the acrylate and formate anions occurred in a complex endothermic process (exp. 28.4%, calc. mass 29.7%). The final product of the thermal decomposition above 630 °C was zinc oxide (exp. 24.5%, calc. mass 24.1%), which was confirmed by the powder XRD analysis (Appendix A).

## 3. Materials and Methods

### 3.1. Materials and Physical Measurements

Melamine (purity > 99%) and DMF (purity ≈ 99.5%) were purchased from Fluka (Saint-Louis, MO, USA) as reagent grades and were used without further purification. Complex [Zn(bpy)(acr)_2_]·H_2_O (**1**) was synthesized as previously reported [24].

Chemical analysis of carbon, nitrogen, and hydrogen was performed using a PE 2400 analyzer (Perkin Elmer, Waltham, MA, USA). The IR spectra were recorded in KBr pellets with a Tensor 37 spectrometer (Bruker, Billerica, MA, USA) in the range 400–4000 cm^−1^. The simultaneous TG/DTA measurement was performed on a Labsys 1200 SETARAM (Caluire-et-Cuire, France) TGA instrument. The sample was placed into alumina crucible and heated from 25 to 800 °C with heating rate of 10 K min^−1^ in an air atmosphere. The flow rate was 16.66 cm^3^ min^−1^ and there was an empty crucible that served as a reference.

X-ray single crystal diffraction datasets were collected with a Bruker D8 Venture Dual Source 100 CMOS diffractometer. The programs used are as follows: data collection: APEX3 V2016.1-0 (Bruker AXS Inc., Madison, Wisconsin, USA, 2016) [41]; cell refinement: SAINT V8.37A (Bruker AXS Inc., 2015) [41]; data reduction: SAINT V8.37A (Bruker AXS Inc., 2015) [41]; absorption correction: SADABS V2014/7 (Bruker AXS Inc., 2014) [41]; structure solution: SHELXT-2015 [42]; structure refinement: SHELXL-2015 [43]; and graphics: XP [44]. R-values are given for observed reflections, and wR2 values are given for all reflections. Exceptions and special features: Compound (**1a**) was refined as an inversion twin (TWIN, BASF). CCDC reference number is 2226321 and contains the supplementary crystallographic data for compounds.

Powder X-ray diffraction was performed using a PROTO AXRD benchtop powder diffractometer (CuKα radiation) (Malvern, United Kingdom).

### 3.2. Syntheses of Precursor and Coordination Polymer

Synthesis of precursor [Zn(bpy)(acr)_2_]·H_2_O (**1**): A suspension containing 0.65 g ZnO and 1.2 mL acrylic acid (ρ = 1.05 g mL^−1^) in 40 mL distilled water was stirred for four hours at room temperature. The reaction mixture was filtered off and to the colorless solution added 20 mL of ethanol containing 1.24 g 2,2′-bipyridine to it. The resulting solution was stirred at room temperature for 2 h until the color became pale yellow. This mixture was allowed to evaporate slowly at room temperature, and after one week, pale yellow crystals began to appear. These were filtered off, washed with ethanol, and air-dried [24].

Synthesis of coordination polymer [Zn(bpy)(acr)(HCOO)]_n_ (**1a**): A total of 0.72 g (2 mmol) [Zn(bpy)(acr)_2_]·H_2_O and 0.26 g (2 mmol) melamine were dissolved in a mixture of methanol, water, and DMF (1:1:1, v/v/v). The obtained solution was continuously stirred at 50 °C for 20 h and then left to slowly evaporate at room temperature. The unreacted melamine crystallized first, and then it was separated by filtration. From the remaining solution, colorless crystals, which were suitable for X-ray measurements, were formed, filtered off, washed several times with cold methanol, and air-dried. The calculated yield of the synthesis process was 56.4%.

The powder diffraction pattern for complex (**1a**) (Appendix A) matched well with that simulated from single crystal structure data, indicating that bulk sample was isolated at pure phase.

### 3.3. X-ray Crystal Structure Analysis of ***1a***

A colorless, plate-like specimen of C_14_H_12_N_2_O_4_Zn, with approximate dimensions of 0.070 mm × 0.150 mm × 0.330 mm, was used for the X-ray crystallographic analysis. The X-ray intensity data were measured on a Bruker D8 Venture PHOTON 100 CMOS system equipped with a micro focus tube Mo ImS (‘Mo Kα’, λ = 0.71073 Å) and a MX mirror monochromator. The integration of the data using an orthorhombic unit cell yielded a total of 4650 reflections at a maximum θ angle of 25.35° (0.83 Å resolution), of which 4650 were independent (average redundancy of 1.000, completeness = 98.4%, and R_sig_ = 2.27%) and 4538 (97.59%) were greater than 2σ(F^2^). The final cell constants of a = 10.1942(1) Å, b = 8.7005(2) Å, c = 31.1599(4) Å, and volume = 2763.72(8) Å^3^ are based upon the refinement of the XYZ centroids of reflections above 20 σ(I). Data were corrected for absorption effects using the multi-scan method (SADABS). The calculated minimum and maximum transmission coefficients (based on crystal size) are 0.5890 and 0.8850. The structure was solved and refined using the Bruker SHELXTL Software Package, using the space group *Pca*2_1_, with Z = 8 for the formula unit, C_14_H_12_N_2_O_4_Zn. The final anisotropic full-matrix least-squares refinement on F^2^ with 380 variables converged at R1 = 3.33% for the observed data and wR2 = 8.11% for all data. The goodness-of-fit score was 1.083. The largest peak in the final difference electron density synthesis was 0.302 e^−^/Å^3^ and the largest hole was −0.454 e^−^/Å^3^ with an RMS deviation of 0.049 e^−^/Å^3^. On the basis of the final model, the calculated density was 1.623 g/cm^3^ and F(000), 1376 e^−^.

## 4. Conclusions and Further Perspectives

The newly synthesized compound [Zn(bpy)(acr)(HCOO)]_n_ (**1a**), which contains two different carboxylate ions, was characterized by a single crystal X-ray diffraction, elemental analysis, IR spectroscopy, and thermal studies. If in the mononuclear precursor, [Zn(bpy)(acr)_2_]·H_2_O, both acrylate ions acted as chelate ligands, in the new afforded complex, acrylate ions act as unidentate. Meanwhile, the formate ions that resulted after DMF hydrolysis are bridged between metallic ions. The stereochemistry of Zn(II) was also changed from octahedral to square pyramidal after its conversion from the mononuclear to polynuclear complex.

On the basis of the reviewed literature data and to our knowledge, the newly synthesized coordination polymer, [Zn(bpy)(acr)(HCOO)]_n_, is the first complex that contains acrylate and other carboxylates in its composition. This result may represent a starting point for further research, which entails DMF hydrolysis and the exploitation of this behavior by finding the optimum reaction conditions (including use of MA) to obtain new complexes with formate, dimethylamine, or, successfully, both of them. On the other hand, the possible catalytic effect of other metallic ions other than Zn(II) ions over DMF hydrolysis will be further investigated.

## Data Availability

Not applicable.

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
