# Peer review of "An Interesting Conversion Route of Mononuclear Zinc Complex to Zinc Mixed Carboxylate Coordination Polymer"

_molecules, 2023, doi:10.3390/molecules28052011_

Round 1

Reviewer 1 Report

The scientific content of the ms. describes an interesting conversion route of mononuclear zinc complex to zinc mixed carboxylate coordination polymer. The authors have isolated and structurally characterize the coordination polymer [Zn(bpy)(acr)(HCOO)]n (bpy = 2,2′-bipyridine, and Hacr = acrylic acid). The characterization of the compound is limited to IR and thermogravimetric analysis. The present study is well-presented and the experimental findings are discussed satisfactorily, yet additional measurements are required. There are some points that should be considered and minor revision of this ms. is required in order to be accepted for publication:

(a) Results, 2.1 Synthesis of the complex:

The authors in Scheme 1 illustrate the reaction of the product’s formation. A more detail illustration of the reaction’s mechanism could be added. How did DMF contributed on the conversion route? Did the authors examine other solvents as well?

(b) Results, 2.3 IR spectra:

In Fig. 2 the legends of the two spectra should be added in order to be clear to readers even in black and white version.

(c) Results, 2.4 Synthesis of the complex:

The stability of this complex in solution would be an important information, given the potential application of the complex. Therefore taking into account that this I a Zinc complex (diamagnetic), the authors could perform 1H NMR analysis in different solvents.  

(d) Supporting Information: I do not feel that Fig. S1 bears any scientific soundness and could be omitted

Reviewer 2 Report

In presented paper a conversion route of mononuclear zinc complex [Zn(bpy)(acr)2]H2O to zinc mixed carboxylate coordination polymer [Zn(bpy)(acr)(HCOO)]n has been reported. The product complex has been characterized by single crystal X-ray diffraction, IR and thermogravimetric analysis. There are several significant shortcomings in this work, out of which I mention following:

1. As follows from the introduction, the hydrolytic decomposition of N,N'- dimethylformamide (DMF) has already been studied (observed) in many previous works.

2. There is not clear, whether or not the presence of melanin (MA) in the reaction mixture is necessary for the reaction to proceed.

3. When preparing the coordination polymer, its yield is not reported (section 3.2). This is a serious deficiency. One cannot rule out that DMF is not contaminated with a small amount of formic acid.

4. The purity of the obtained coordination polymer is not guaranteed as it was not checked by X-ray powder diffraction.

5. The prepared coordination polymer does not appear to possess any interesting properties or uses.

Based on these comments, I propose to reject presented work. 

Reviewer 3 Report

1. “..The novelty of current research is sustained by that so far, to our knowledge…” I still not suggest the authors used the word “novel” in the full text, all of the work is only described as new. NOT Novel.

2. Source and purity of all chemicals used should be specified in the experimental section.

3. Table 1 should be removed into ESI.

4. I suggest the authors discuss the weak interactions between the adjacent chains.

5. In the introduction section, the authors should rewrite this part, each part should have a main topic, not only illustrate this work, you should highlight the full design of the work?

6.  “As it is known, the Δ value (Δ = νas(COO) - νs(COO)) can be associated with the coordination mode of carboxylate group. Therefore, for complex (1a), a Δ value of 253 cm-1 resulted from bands located at 1619 and 1366 cm-1 indicates a unidentate coordination for acrylate,” This part should be highlighted the updated document, such as CrystEngComm, 2022, 24, 6933–6943 and Micropor. Mesopor. Mat, 341(2022) 112098. Also, in this part, It  should be noted that the acrylate ion present an asymmetric chelating mode of the 98 carboxylate group O1/O2, as indicated by the Zn – O bond values (Zn1 – O1 2.076 Å and 99 Zn1 – O2 2.562 Å), with a difference of ca. 0.48 A. This should be compared the values, such as Inorganics, 10(2022) 202 and CrystEngComm, 2017,19:4368-4377.

7. The final product of the thermal decomposition above 630 °C is zinc oxide (exp. 24.5%, calc. mass 136 24.1%), which was confirmed by powder XRD analysis. Please give the XRD data for support.

8. “The formate ion resulted after hydrolysis act as bridging ligand leading to a polynuclear complex, this being out of the expectation of our original purpose “ I suggest the authors list a Table and conclude the different condition for the hydrolysis of DMF.

9. The manuscript contains spelling/grammatical errors. So, the language should be polished thoroughly. Such as …” It shoud be noted that”

Round 2

Reviewer 2 Report

I am satisfied with the answers of the authors, therefore I recommend accepting the article after minor revision.

Reviewer 3 Report

it is nice for me.